# Hair-Follicle-Associated Pluripotent (HAP) Stem Cells Can Extensively Differentiate to Tyrosine-Hydroxylase-Expressing Dopamine-Secreting Neurons

**DOI:** 10.3390/cells10040864

**Published:** 2021-04-10

**Authors:** Michiko Yamane, Nanako Takaoka, Koya Obara, Kyoumi Shirai, Ryoichi Aki, Yuko Hamada, Nobuko Arakawa, Robert M. Hoffman, Yasuyuki Amoh

**Affiliations:** 1Department of Dermatology, Kitasato University School of Medicine, Minami Ward, Sagamihara 252-0374, Japan; mm20054@st.kitasato-u.ac.jp (M.Y.); toukyoutoootakukamiikedai@yahoo.co.jp (N.T.); obarakoya@yahoo.co.jp (K.O.); kyohya@med.kitasato-u.ac.jp (K.S.); ryoaki@med.kitasato-u.ac.jp (R.A.); hyuko@kitasato-u.ac.jp (Y.H.); hql04171@gmail.com (N.A.); 2AntiCancer, Inc., 7917 Ostrow Street, San Diego, CA 92111, USA; 3Department of Surgery, University of California, San Diego, CA 92037-7220, USA

**Keywords:** hair follicle, stem cell, hair follicle stem cell area, differentiation, neuron, dopamine, neurogenesis, neural stem cells

## Abstract

Hair-follicle-associated pluripotent (HAP) stem cells are located in the bulge area of hair follicles from mice and humans and have been shown to differentiate to neurons, glia, keratinocytes, smooth muscle cells, melanocytes and beating cardiac muscle cells in vitro. Subsequently, we demonstrated that HAP stem cells could effect nerve and spinal-cord regeneration in mouse models, differentiating to Schwann cells and neurons in this process. HAP stem cells can be banked by cryopreservation and preserve their ability to differentiate. In the present study, we demonstrated that mouse HAP stem cells cultured in neural-induction medium can extensively differentiate to dopaminergic neurons, which express tyrosine hydroxylase and secrete dopamine. These results indicate that the dopaminergic neurons differentiated from HAP stem cells may be useful in the future to improve the symptoms of Parkinson’s disease in the clinic.

## 1. Introduction

Parkinson’s disease is progressive and decreases the quality of life of the patient over time, and is often causing death. Standard of care is a carbidopa and levodopa combination, which often poorly control the symptoms of Parkinson’s disease. Cell transplantation into Parkinson’s disease patients to replace the lost dopaminergic neurons of the substantia nigra pars compacta has been studied since the 1980s.

Hair-follicle-associated pluripotent (HAP) stem cells are located in the hair-follicle bulge area and can differentiate into neurons, glia, keratinocytes, smooth muscle cells, melanocytes, and beating cardiac muscle cells [1,2,3,4,5]. HAP stem cells express the neural stem-cell marker nestin and the embryonic stem-cell markers Nanog and Oct4 [6].

HAP stem cells from mice have been used to repair the severed sciatic nerve in mouse models [7,8,9,10,11,12]. The implanted HAP stem cells differentiated into Schwann cells in the re-joined nerve and restored nerve and leg function. Human HAP stem cells, which have a similar differentiation potential as mouse HAP stem cells, have also been used to restore the structure and function of the severed sciatic nerve in mice [4,5,13]. HAP stem cells have been used to repair the severed spinal cord in mice, leading to improved hindlimb locomotion [8,14]. The implanted HAP stem cells in the re-joined spinal cord differentiated to oligodendrocytes and βIII-tubulin-positive neuron-like cells [15].

In the present study, we demonstrate that mouse HAP stem cells can extensively differentiate to tyrosine-hydroxylase expressing, dopamine-secreting neurons. The dopaminergic neurons differentiated from HAP stem cells have future potential to improve the symptoms of Parkinson’s disease, as they can be readily isolated and banked from everyone.

## 2. Materials and Methods

### 2.1. C57BL/6 Mice

Six weeks old C57BL/6J female mice (20 g or more) (CLEA Japan, Tokyo, Japan) were used to isolate vibrissa hair follicles [1,2]. The experimental animals were housed at 24 ± 1 °C, relative humidity of 50–60%, and 14 h of light and 10 h of dark intervals. 

### 2.2. Isolation of Vibrissa Hair Follicles and Induction of Dopaminergic Neurons from HAP Stem Cells In Vitro

Vibrissa hair follicles from mice were isolated as previously reported [16]: To isolate the vibrissa hair follicles from C57BL/6 mice, the animals were anesthetized with a combination anesthetic of 0.75 mg/kg medetomidine, 4.0 mg/kg midazolam and 5.0 mg/kg butorphanol [17]. Their upper lip containing the vibrissa pad was cut, and the inner surface was exposed. Intact vibrissa hair follicles were dissected under a binocular microscope and plucked from the pad by pulling them gently by the neck with fine forceps. All surgical procedures were performed in a sterile environment. The upper part of vibrissa hair follicles was separated under a binocular microscope. HAP stem cells from the isolated upper part of the vibrissa hair follicle were initially cultured in fresh DMEM (Sigma, St. Louis, MO, USA) containing 10% fetal bovine serum (FBS), 50 µg/mL gentamicin (GIBCO, Grand Island, NY, USA), 2 mM L-glutamine (GIBCO), 10 mM HEPES (MP Biomedicals, Santa Ana, CA, USA) for 7 days [16]. For differentiation to dopaminergic neurons, HAP stem cells growing from the upper part of vibrissa hair follicle were switched to neural-induction medium (STEMdiff Dopaminergic Neuron Differentiation Kit, STEMCELL Technologies, Vancouver, BC, Canada) containing neural-progenitor-cells-induction medium, dopaminergic-neuron-differentiation medium, dopaminergic-neuron-maturation medium-1, and dopaminergic-neuron-maturation medium-2 and cultured according to the instructions of the manufactures (Figure 1). As a control, HAP stem cells were cultured in non-induction medium (10% FBS DMEM). The cells differentiated from HAP stem cells were prepared for immunofluorescence staining, FACS, and for measurement of dopamine secretion. 

### 2.3. Immunofluorescence Staining

Immunofluorescence staining of differentiated cells was performed as previously reported [2]: The primary antibodies used were anti-βIII-tubulin mouse monoclonal (1:500, MMS-435P, Tuj1 clone, Covance, Princeton, NJ, USA); anti-tyrosine-hydroxylase (TH) rabbit polyclonal (1:100, NB300-109, Novus Biologicals, Centennial, CO, USA); anti-dopamine polyclonal antibody (1:250, IS1005, ImmuSmol, Pessac, France) with a STAINperfect immnostaining kit A (SP A-1000, ImmuSmol); anti-dopamine-transporter (DAT) rabbit monoclonal antibody (1:250, ab184451, Abcam, Cambridge, UK); and anti-Nurr1 rabbit polyclonal (1:50, 10975-2-AP, Proteintech, Rosemont, IL, USA). The secondary antibodies were Alexa Fluor^®^ 568-conjugated goat anti-mouse (1:400, A11004, Molecular Probes, Eugene, OR, USA); Alexa Fluor^®^ 488-conjugated goat anti-rabbit (1:400, A11008, Molecular Probes); and Alexa Fluor^®^ 568-conjugated goat anti-rabbit (1:400, A21069, Molecular Probes). Counterstaining was performed with 4′, 6-diamino-2-phenylindole, dihydrochloride (DAPI) (SE196, DOJINDO, Kumamoto, Japan). Images were collected using an LSM 710 microscope System with ZEN software (Carl Zeiss, Oberkochen, Germany).

### 2.4. Fluorescence-Activated Cell Sorting (FACS)

FACS was performed as previously reported [1,2]: The primary antibodies used were anti-tyrosine-hydroxylase (TH) monoclonal antibody (1:100, ab129991, Abcam) and anti-βIII-tubulin mouse monoclonal antibody (1:500) The secondary antibodies used were goat anti-mouse IgG H&L phycoerythrin (1:500, ab97041, Abcam) and Alexa Fluor^®^ 488-conjugated goat anti-mouse (1:500, A11001, Molecular Probes). The cells were analyzed by FACS Verse (BD Bioscience, San Jose, CA, USA), using FACS suiteTM software (BD Bioscience). FACS analyses were repeated in triplicate.

### 2.5. High Performance Liquid Chromatography (HPLC)

Dopamine was analyzed using HPLC as previously reported [18]: HAP stem cells that differentiated to dopaminergic neurons were lysed with PCA buffer (Perchloric acid 0.2 M, EDTA-2Na 100 μM), then measured by HPLC, using a TSK gel ODS-80TM column (TOSOH BIOSCIENCE, Tokyo, Japan) and an electrochemical detector system EDC-100 (EICOM, Kyoto, Japan).

### 2.6. Ca^2+^ Imaging

Ca^2+^ imaging was performed on cells grown in 35 mm glass bottom microwell dishes (MatTek, Ashland, MA, USA) with the calcium-sensitive dye Fluo 4-AM (DOJINDO) 1 μM and AM ester-dissolving reagent Pluronic F-127 (0.04%) (FUJIFILM Wako, Osaka, Japan) in HEPES buffer (NaCl 145 mM; MgCl_2_ 1 mM; KCl 5 mM; glucose 5.5 mM; CaCl_2_ 1 mM; HEPES 10 mM; pH 7.4) [2]. Fluo-4 fluorescence images (488 nm excitation) were collected and recorded at 100 frames. After 10 s of image acquisition, ATP was added. We examined real-time movie files of continuously recorded data to assess changes in cell fluorescence that occur in response to ATP stimulation. Fluorescence images were collected and recorded using an LSM 710 microscope System with ZEN software (Carl Zeiss, Oberkochen, Germany).

### 2.7. Statistical Analysis

The experimental data are expressed as the mean ± SD. Statistical analyses were performed with the unpaired Student’s t-test. A probability (P) value of *p* ≤ 0.05 was considered significant.

## 3. Results

### 3.1. HAP Stem Cells Differentiate Efficiently to Dopaminergic Neurons

HAP stem cells were cultured in neural-induction medium for 45 days. As a control, HAP stem cells were cultured in non-induction medium for 28 days. HAP stem cell differentiated to βIII tubulin-positive, tyrosine-hydroxylase-positive cells, which secreted dopamine by day 7 after switching to neural-induction medium, indicating they were dopaminergic neurons (Figure 1). Tyrosine-hydroxylase, DAT and Nurr1 in the differentiated dopaminergic neurons were observed by immunofluorescence staining (Figure 2). Fluorescence-activated cell sorting (FACS) analysis showed that the percentage of cells differentiating to dopaminergic neurons in neural-induction medium was 48.90 ± 4.64% compared to 15.53 ± 7.47% in non-induction medium (*p* = 0.0014) (Figure 3A). The percentage of cells differentiating to βIII tubulin-positive neurons in neural-induction medium was 62.64% (Figure 3A) and in non-induction medium it was 15.4% [1].

### 3.2. HAP Stem Cells Differentiated to Dopaminergic Neurons That Extensively Proliferate

HAP stem cells, differentiated to dopaminergic neurons, extensively proliferated in neural-induction medium to 6.67 ± 3.06 × 10^3^ cells/hair follicle. 

### 3.3. HAP Stem Cells Differentiated to Dopaminergic Neurons Secreted Dopamine at High Levels

HPLC analysis showed that HAP stem cells differentiated to dopaminergic neurons that secreted dopamine at 3.73 ± 0.41 ng/hair follicle and 3,4-dihydroxyphenylacetic acid (DOPAC) at 1.96 ± 0.04 ng/hair follicle in neural-induction medium. Significantly more dopamine was secreted in neural-induction medium compared to non-induction medium (*p* = 0.0002) (Figure 3B,C). 

### 3.4. HAP Stem Cells Differentiated to Dopaminergic Neurons Have Increased Ca^2+^ Levels When Treated with ATP

HAP stem cells differentiated to dopaminergic neurons increased their Ca^2+^ levels when treated with 300 μM ATP (Figure 4, Appendix A).

## 4. Discussion

In 2020 a case report was published on a Parkinson’s patient transplanted with dopamine-producing iPSC derived from the patient’s skin cells. The dopamine-producing iPSC were transplanted into the putamen of the patient who showed stability and improvement [19]. However, it took seven steps to produce the dopamine-producing iPSC, including forced miRNAs expression, transfect with reprogramming factors, and chemical elimination of potential tumorigenic un-differentiated cells [20]. None of these genetic steps to induce differentiation and chemical treatment to remove potential tumorigenic cells are necessary with dopamine-producing HAP stem cells.

The present report demonstrates that HAP stem cells can efficiently differentiate to dopaminergic neurons, which proliferate well and secrete high amounts of dopamine. Furthermore, HAP stem cells matured as neurons when cultured in neural induction medium demonstrated by increased levels of Ca^2+^ upon treatment with ATP. Future studies need to shorten the culture period and increase the differentiation rate of dopamine-producing neurons.

Doi et al. [21] and Kikuchi et al. [22] demonstrated that implanted human induced pluripotent-stem-cell (hiPSC)-derived dopaminergic progenitor cells improved the motor behavior of 6-OHDA lesioned rats. Narytnyk et al. [23] demonstrated that human epidermal neural-crest stem cells could form dopaminergic neurons. Alizadeh et al. [24] demonstrated that when cultured olfactory bulb neural stem cells, from brain-dead donors, were supplemented with sonic hedgehog, fibroblast growth factor-8, and glial-derived neurotrophic factor, they differentiated to dopaminergic neurons. The differentiated dopaminergic neurons expressed dopaminergic markers tyrosine-hydroxylase and aromatic L-amino acid decarboxylase (AADC). Nakagawa et al. [25] demonstrated that human embryonic stem cells and hiPSC, cultured in xeno-free medium, could differentiate to dopaminergic neurons. Hartfield et al. [26] demonstrated that hiPSC differentiated to mature substantia nigra pars compacta dopaminergic neurons. Human iPSC-derived dopaminergic progenitor cells transplanted to the putamen of *Macaca fascicularis* monkeys that had Parkinson’s symptoms alter neurotoxic treatment, improved their spontaneous movement. The transplantation required immunosuppression [22]. 4.0 − 8.0 × 10^4^ TH positive neurons in the human brain may be required to achieve a meaningful therapeutic effect [27].

HAP stem cells, located in the hair follicle bulge area, used in the present study, are the most accessible stem cells compared to other stem cell types. As shown in the present report, HAP stem cells differentiated efficiently to dopaminergic neurons that extensively proliferated, expressed tyrosine-hydroxylase and secreted large amounts of dopamine without genetic manipulations or added growth factors, and do not form tumors [8]. Furthermore, human HAP stem cells differentiated into 5.0 ± 1.7 × 10^4^ cells/hair follicle for 4 weeks [5]. HAP stem cells, which were discovered by Li et al [28], may be most useful in the future to improve the symptoms of Parkinson’s disease. The possibility of clinical use of HAP stem cells for Parkinson’s disease is feasible and practical, since HAP stem cells are readily available from everyone and can be cryopreserved and banked without loss of pluripotency [5,16].

## Figures and Tables

**Figure 1 cells-10-00864-f001:**
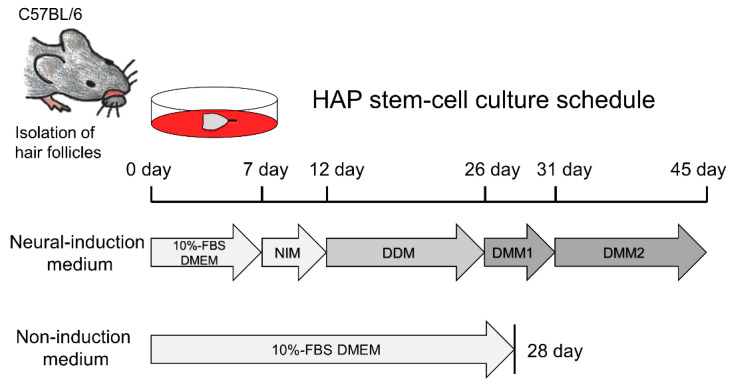
Scheme for differentiation of dopaminergic neurons from HAP stem cells. HAP stem cells were initially cultured in neural-induction medium, which was changed as follows: Day 7, neural-progenitor-cells-induction medium (NIM); Day 12, dopaminergic-neuron-differentiation medium (DDM); Day 26, dopaminergic-neuron-maturation medium-1 (DMM1); Day 31, dopaminergic- neuron-maturation medium-2 (DMM2).

**Figure 2 cells-10-00864-f002:**
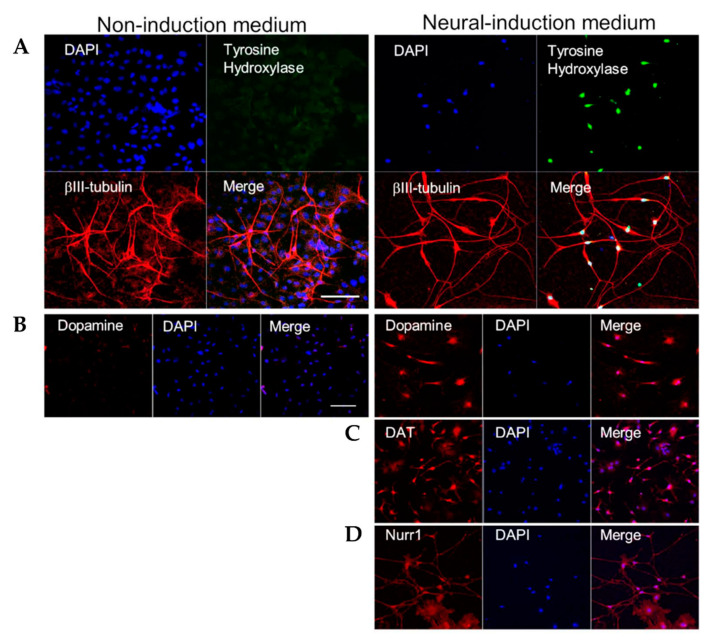
HAP stem cells differentiated to dopaminergic neurons. The left panels show HAP stem cells cultured in non-induction medium. The right panels show HAP stem cells cultured in neural-induction medium. (**A**) Immunofluorescence staining shows that HAP stem cells cultured in neural-induction medium differentiated to βIII-tubulin-positive (red fluorescence) and tyrosine-hydroxylase-positive (green fluorescence) neurons. (blue fluorescence = DAPI). Tyrosine-hydrox-ylase was expressed much more strongly in neural-induction medium. (**B**) HAP stem cells cultured in neural-induction medium differentiated to dopamine-positive (red fluorescence) dopaminergic neurons. Dopamine was produced much more in neural-induction medium. (**C**) HAP stem cells cultured in neural-induction medium differentiated to DAT positive (red fluorescence). (**D**) HAP stem cells cultured in neural-induction medium differentiated to Nurr1 positive (red fluorescence). Scale bar = 100 μm.

**Figure 3 cells-10-00864-f003:**
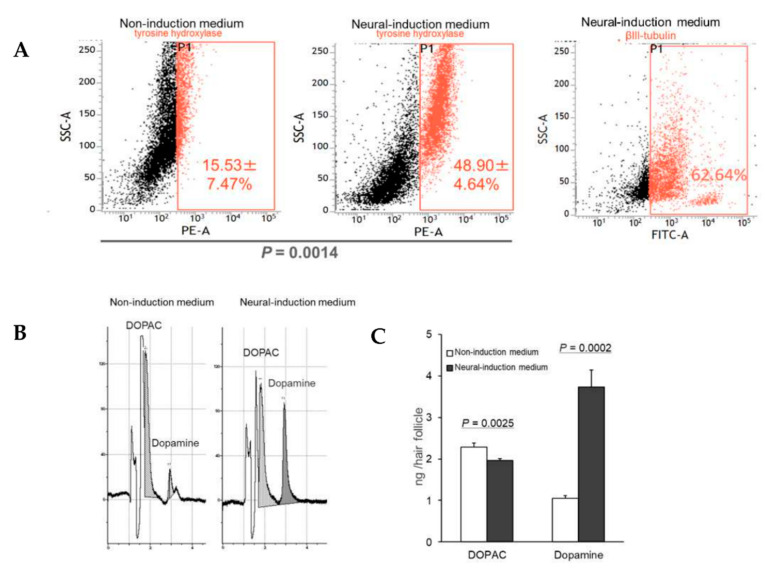
Proliferation of dopaminergic neurons differentiated from HAP stem cells. (**A**) FACS analysis showed that the dopaminergic neurons differentiated efficiently in neural-induction medium. (PE-A = tyrosine-hydroxylase-phycoerythrin. FATC-A = βIII-tubulin). Dopaminergic neurons differentiated from HAP stem cells secreted high levels of dopamine when cultured in neural-induction medium. (**B**) HPLC shows that the dopaminergic neurons secreted large amounts of dopamine. (**C**) Dopamine secretion significantly increased in neural-induction medium compared to non-induction medium. DOPAC = 3,4-dihydroxyphenylacetic acid.

**Figure 4 cells-10-00864-f004:**
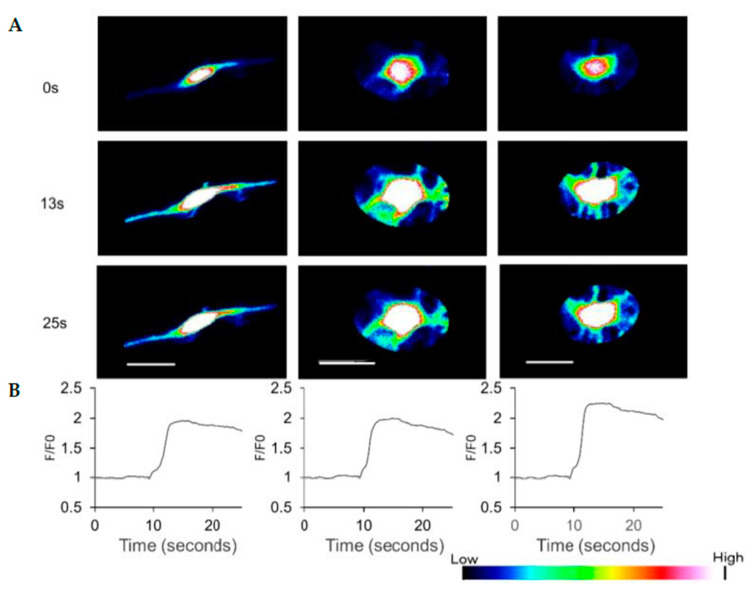
Effect of ATP on Ca^2+^ levels in dopaminergic neurons differentiated from HAP stem cells. (**A**) ATP caused Ca^2+^ concentration changes in the dopaminergic neurons with time observed. (**B**) The line-plots in the bottom panels show the relative fluorescence ratio (F/F0), where F0 is fluorescence before treatment with ATP, and F is fluorescence after treatment. Scale bar = 20 μm.

## Data Availability

The data presented in this study are available on reasonable request from M.Y. (mm20054@st.kitasato-u.ac.jp).

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
