# Peer review of "Hair-Follicle-Associated Pluripotent (HAP) Stem Cells Can Extensively Differentiate to Tyrosine-Hydroxylase-Expressing Dopamine-Secreting Neurons"

_cells, 2021, doi:10.3390/cells10040864_

Round 1

Reviewer 1 Report

The article "Hair-follicle-associated pluripotent (HAP) stem cells can extensively differentiate to tyrosine-hydroxylase-expressing dopamine-secreting neurons" by Yamane et al. presents novel data which are of interest to the scientific community, even for their potential clinical application. However, the paper should be extensively revised, both in its format and in its content, before acceptance by the Journal. 

Major points:

The Introduction section is not particularly informative, containing the same synthetic background information, as in the Abstract. It should be extended, for example providing details on PD model.

The Authors state: "HAP stem cells from the isolated upper part of vibrissa hair follicle were initially cultured ..." How were HAP stem cells characterized? Was any specific assay performed?

The Discussion section needs extensive revision. In the present form, it reports other groups' work in a list, rather than critically discussing their data with Authors' results. Interpretation of data presented in this article should be instead done in view of the available literature, in a comprehensive manner.

Minor points:

In the Materials and Methods section, all the protocols refer to previous studies. I feel that at least major information on the procedures should be given. Also, information on chemicals and reagents are incomplete in most cases. For example, all antibody product codes and company names of several reagents are missing.

References are listed in a inhomogenous format, journals are sometimed abridged, somentimes reported in extenso. I think also links to the websites (sometimes active) should be deleted.

Throughout the text, words are unnecessarily divided by a "-", please correct these typographycal errors.

Author Response

Response to Reviewer 1 

Thank you for your review of our paper. We have answered each of your points in the attached file.

Reviewer 2 Report

This study shows the direct differentiation of hair-follicle associated stem cells (HAP) into neurons. The authors could demonstrate with different methods that they have TH positive cells in their culture and under neuronal-induction medium they release dopamine. The experimental process is clearly structured and easy to understand. I recommend this article for publication, but a few additions should be done during revision.

  1. The authors state that the use of dopaminergic neurons from HAP cells may useful to improve the symptoms of Parkinson’s disease. This statement is a bit too vague, please explain in more detail.
  2. Figure 2: Could you please improve the immunofluorescence stainings to get a better impression of the cells.
  3. Could the authors please comment on the fact, that day 28 was chosen for control experiments/non-neural induction medium and day 45 for neural-induction medium. Why was not the same time chosen for the experiments?
  4. What is the overall percentage of neurons? Which other neural and non-neural cell types are generated within the differentiation? Please provide additional information/experiments to answer these questions.
  5. The expression for other markers which are related to dopaminergic neurons, e.g. DAT, AADC, NURR1 should be checked.
  6. Could the authors please state that only TH positive neurons have been used for Ca analysis. Please give more details about the experimental setup/controls to verify the usage of dopaminergic neurons and not other cells within in the differentiation.

Minor comments:

  1. There are many words which are incorrectly separated by a hyphen e.g. se-crete (21), hair-follicle (28), im-planted (38)… please check the whole manuscript for such typing errors
  2. Typo (18): “stem cells are can be banked”
  3. Line break (84), (96)
  4. Check for wrong symbols: ßIII tubulin (118), 300µM ATP (153)

Author Response

Response to Reviewer 2 

Thank you for your review of our paper. We have answered each of your points in the attached file.

Round 2

Reviewer 1 Report

I can see some good work has been done by the Authors to improve the quality of the manuscript. However, I still find the text too synthetic, particularly the Discussion section.

Author Response

Thank you for your review of our paper. We have answered each of your points in the attached file.

Reviewer 2 Report

The authors have accomplished the required improvements.

The manuscript can be accepted in present form.

Author Response

(The authors gave the same response as above.)
